# Continuous transcriptome analysis reveals novel patterns of early gene expression in *Drosophila* embryos

## Graphical abstract

## Authors

J. Eduardo Pérez-Mojica, Lennart Enders, Joseph Walsh, Kin H. Lau, Adelheid Lempradl

## Correspondence

heidi.lempradl@vai.org

## In brief

Pérez-Mojica et al. have developed an operationally simple RNA sequencing approach for enabling high-resolution, time-sensitive transcriptome analysis in early-stage *Drosophila* embryos (0–3 h). Their analysis provides detailed insights into gene expression during early development and enhances the current understanding of the earliest sex-specific transcriptional signatures.

## Highlights

- An accessible RNA sequencing methodology for single *Drosophila* embryos

- A high-resolution, time-sensitive transcription dataset for early *Drosophila* embryos

- Identification of the earliest evidence of sex-biased transcription

- Dynamic expression patterns for thousands of genes during early development

Pérez-Mojica et al., 2023, Cell Genomics 3, 100265
March 8, 2023 © 2023 The Author(s).

# Cell Genomics

CellPress

## Technology

# Continuous transcriptome analysis reveals novel patterns of early gene expression in *Drosophila* embryos

J. Eduardo Pérez-Mojica,[1] Lennart Enders,[2,4] Joseph Walsh,[1] Kin H. Lau,[3] and Adelheid Lempradl[1,2,5,*]

[1]Department of Metabolic and Nutritional Programming, Van Andel Institute, Grand Rapids, MI 4930, USA
[2]Department of Epigenetics, Max Planck Institute of Immunobiology and Epigenetics, 79108 Freiburg im Breisgau, Germany
[3]Bioinformatics and Biostatistics Core, Van Andel Institute, Grand Rapids, MI 4930, USA
[4]Present address: CeMM Research Center for Molecular Medicine of the Austrian Academy of Sciences, 1090 Vienna, Austria
[5]Lead contact
*Correspondence: heidi.lempradl@vai.org

## SUMMARY

The transformative events during early organismal development lay the foundation for body formation and long-term phenotype. The rapid progression of events and the limited material available present major barriers to studying these earliest stages of development. Herein, we report an operationally simple RNA sequencing approach for high-resolution, time-sensitive transcriptome analysis in early (≤3 h) *Drosophila* embryos. This method does not require embryo staging but relies on single-embryo RNA sequencing and transcriptome ordering along a developmental trajectory (pseudo-time). The resulting high-resolution, time-sensitive mRNA expression profiles reveal the exact onset of transcription and degradation for thousands of transcripts. Further, using sex-specific transcription signatures, embryos can be sexed directly, eliminating the need for Y chromosome genotyping and revealing patterns of sex-biased transcription from the beginning of zygotic transcription. Our data provide an unparalleled resolution of gene expression during early development and enhance the current understanding of early transcriptional processes.

## INTRODUCTION

In many animal species, the zygote relies on maternally deposited transcripts to progress through the earliest stages of development.[1] It is not until later that the zygote takes control of its own development, a process referred to as the maternal-to-zygotic transition. An important element of this transition is the start of transcription from the zygotic genome, also referred to as zygotic genome activation (ZGA). Previous studies have shown that the ZGA is a progressive event. It starts with the transcription of just a handful of genes (minor ZGA) and increases to hundreds of genes shortly thereafter (major ZGA).[2–4]

In *Drosophila melanogaster*, it is generally accepted that *sisterless A* (*sis A*) and *snail* (*sna*) are transcribed early in development during nuclear cycle (NC) 8.[5,6] Evidence suggests that *scute* (*sc*) is an additional early transcribed gene, but reports differ on the timing of transcriptional onset.[5,7–9] The total number of zygotic transcribed genes reported to be expressed by NC 9 is 20 and increases to 63 by the end of NC 10.[4] This so-called minor wave of transcription coincides with other important developmental processes, such as the migration of nuclei to the posterior pole of the embryo and the generation of pole cells.[10] With the onset of the minor ZGA, nuclear cycles become progressively longer. While the first eight NCs on average take 8 min, the duration continuously increases until it reaches 65 min at

NC 14.[10] NC 14 marks the beginning of the major ZGA, and the number of genes transcribed increases significantly to 3,540.[4] The major ZGA is accompanied by important developmental processes such as cellularization, first gastrulation movements, end of synchronous nuclear divisions, and the start of dosage compensation by Male-Specific Lethal (MSL) complex.[2,11,12] Of note, sex-specific transcription is observed as early as NC 11.[2] The exact onset and sequence of transcriptional events leading up to this differential gene expression remain poorly understood.

In parallel to ZGA, maternal transcripts are being degraded in an organized manner. This process of clearing the maternally deposited mRNAs is essential for proper development. It is important to note that the degradation of certain maternal transcripts occurs even in unfertilized eggs.[13]

### Design

The limited size of embryos and rapid progression of developmental processes make a quantitative assessment of transcriptional events challenging. Previous studies have investigated the timing, extent, and sex specificity of the ZGA using different methods such as RNA radioactive[14] or metabolic labeling,[4] *in situ* hybridization,[15,16] RNA sequencing (RNA-seq),[2,17] qPCR-based experiments,[9] and direct count of mRNA molecules.[3,18] Notably, all the different methods rely on the staging

of embryos, which can be time consuming, error prone, and requires a well-optimized protocol to guarantee the fast collection of embryos when working with fresh samples. Currently, the only available option to avoid manual embryo staging is to rely on short egg-laying times and incubation to the desired developmental stages. The results of these studies, however, can be biased due to the rate of egg fertilization, regularity of oviposition, and embryo withholding. The latter has been shown to differ for more than 10 h in some *Drosophila* species.[19] The technical difficulties of existing protocols have led to inconsistencies between findings from different laboratories. For instance, more than half of the transcripts assigned to the minor ZGA in one study[9] were likely due to the contamination of a sample with an older embryo.[4]

To address these technical limitations and ensure increased data reproducibility, we developed a single-embryo RNA-seq method to measure zygotic transcripts on a continuous time-scale. Using an analysis pipeline designed for single-cell RNA-seq, we utilize the transcriptome to determine the biological age and sex for each embryo, eliminating human and technical errors introduced by visual staging. The data produced using this method can be corroborated through comparison with published data and provide the first continuous timeline of transcript levels during early development ($\leq$3 h) in *D. melanogaster*.

## RESULTS

### Single-embryo RNA-seq

In order to study early embryonic transcription in a continuous manner, we performed single-embryo RNA-seq on a total of 192 embryos. The embryos were collected from two different cages in three consecutive 1-h time intervals and allowed to develop further for 0, 1, or 2 h. This resulted in an approximate collection time window of 10 min to 3 h.

RNA was isolated from individual embryos to perform single-embryo RNA-seq using a modified CEL-Seq22 protocol.[20,21] The sequencing data were analyzed using the RaceID/StemID/FateID single-cell analysis tool (Figure 1A).[22] Embryos with less than 250,000 reads were excluded from the analysis, leaving 122 embryos for final analysis. In total, we identified 9,777 genes with $\geq$3 unique molecular identifier (UMI) corrected read counts in $\geq$5 embryos. Unsupervised *k*-medoids clustering of our data, according to transcriptome similarities, resulted in 14 clusters (Figure 1B).

Dimensionality reduction of the single-embryo RNA-seq data using t-distributed stochastic neighbor embedding (t-SNE) produced a map where all samples assembled in a linear pattern (Figure 1B). A similar arrangement was confirmed by other dimensionality reduction methods, including a Fruchterman-Reingold force directed layout and principal-component analysis (PCA) (Figures S1A and S1B).

We then used StemID2, an algorithm developed for the derivation of differentiation trajectories in single-cell data, to generate a lineage tree object, where each embryo receives a relative coordinate on the inferred inter-cluster links according to their transcriptome. The ordering of embryos along this computed developmental trajectory is also called pseudo-time. Because mated females can lay unfertilized eggs, which would compro-

mise our analysis, we identified and excluded such embryos based on their position in the pseudo-timeline and the lack of expression of previously reported early transcribed genes (e.g., *scw*, *sc*, and *esg*). At the beginning of development, mRNA decay is the dominating process determining the pseudo-temporal ordering of samples in our analysis. It was previously reported that the decay of certain maternally deposited transcripts occurs even in unfertilized eggs[13]; however, only fertilized eggs will start transcribing genes. This allows us to identify and exclude samples that, due to mRNA decay patterns, are assigned a high pseudo-time but lack expression of these early expressed genes (Figures S1C–S1E). The number of unfertilized eggs (n = 5) in our dataset matches the actual fertility rate measured on the same day as sample collection (Figure S1F). Analyzing only fertilized embryos resulted in a layout similar to the one observed for all samples (117 embryos, Figure 1C), and both t-SNE and pseudo-time analysis show that there is no cage batch effect in either analysis (Figures S1G and S1H).

For our purposes, pseudo-time encompasses the time it takes from the earliest embryo measurements, 10 min after fertilization (i.e., a 10-min delay is necessitated by sample processing time), to the developmental stage represented by the last embryo on the spanning tree (Figure 1D). We subsequently compared this computationally derived pseudo-timeline with our three sample collection time intervals. Even though embryos were collected in three defined 1-h intervals, their position in pseudo-time was not restricted to their respective collection time windows (Figure 1E; Table 1). These results confirm the propensity of mated females to withhold fertilized eggs for extended periods of time. To avoid contamination of samples by withheld embryos, published related methods currently rely on hand staging the developing embryos. Our method enables us to identify these withheld embryos and to assign them to their correct developmental times without the use of labeling protocols or the need to hand stage the embryos. Together, these results show that we have successfully developed a single-embryo RNA-seq protocol and analysis pipeline for characterizing the developmental progress within single embryos.

### Single-embryo sequencing and pseudo-time analysis allow for the continuous assessment of transcription profiles during early embryogenesis

Next, to investigate the expression of genes across early development, we focused our analysis only on embryos up to approximately 3 h of age (84 embryos). This analysis revealed nine stable clusters of embryos based on their transcription (Figures 2A and 2B) and provided a more refined look at the developmental trajectory. To determine if the computationally derived pseudo-time was in agreement with the biological age of the embryos, we assessed the expression of genes that have been previously reported to be activated during the minor[4] or major[3] wave of ZGA (Table S1). Plotting the combined expression of these genes onto our two-dimensional layout (Figures 2C and 2D) and along the pseudo-timeline (Figures 2E and 2F) reveals that these two major transcriptional events coincide with the increased distance in our t-SNE map between clusters 1 and 2 and 4 and 5. This is expected as gaps are expected

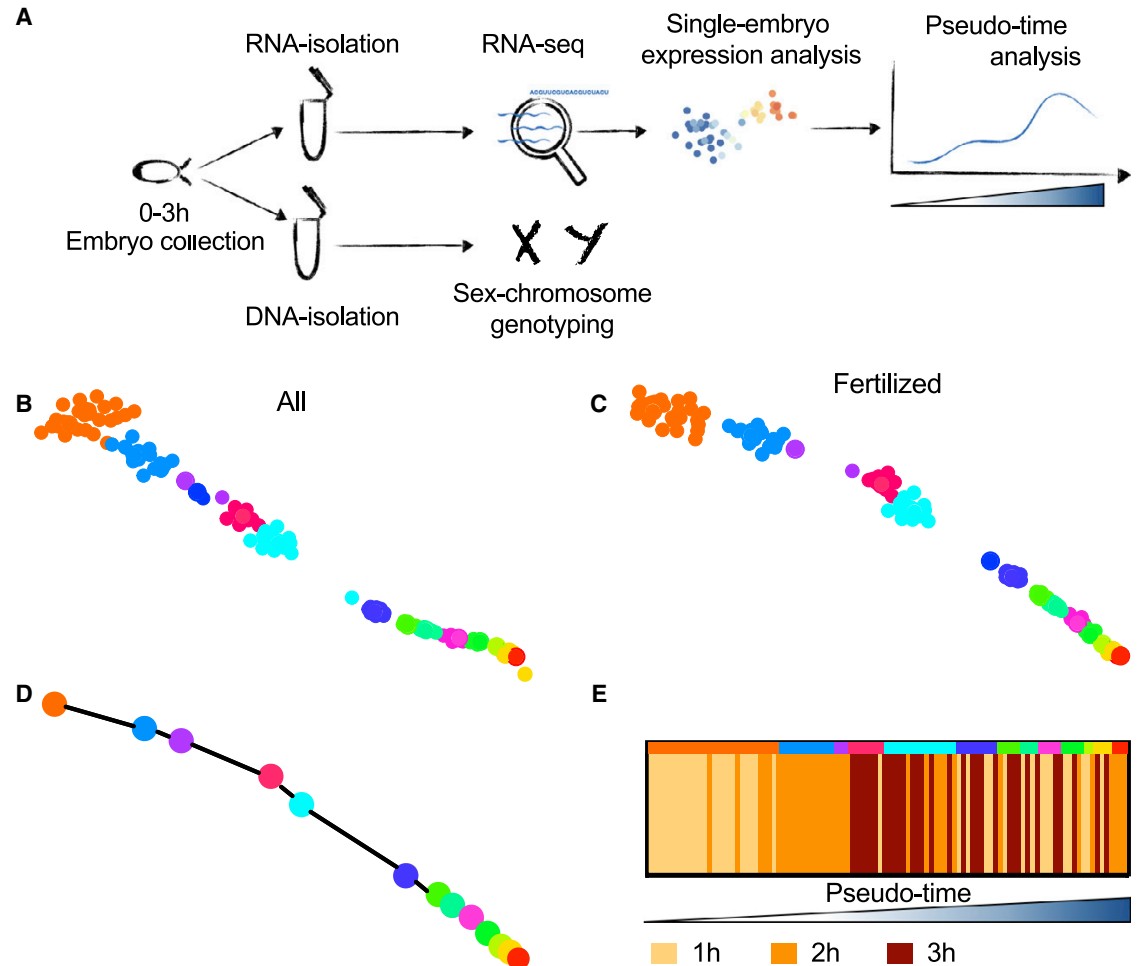

**Figure 1. Single-embryo RNA-seq approach to bioinformatically identify developmental age**

(A) Schematics of methodology: single embryos are collected in 1-h intervals and aged up to 3 h. RNA and DNA are isolated from the same single embryos. DNA is used for genotyping the X and Y chromosomes, while RNA is processed using a modified CEL-Seq2 protocol to determine embryo age.

(B and C) (B) t-SNE before (n = 122) and (C) after the removal of unfertilized eggs (n = 117) with clusters identified by k-medoids clustering indicated by different colors.

(D) Lineage analysis by StemID2/FateID identified a single trajectory for all clusters (with n > 1 embryos) resulting in the ordering of embryos along a pseudo-time axis according to their age.

(E) Comparison of the pseudo-time order with the actual collection time intervals. Ascending pseudo-time (embryo age) from left to right, colors in top bar indicate clusters from (C).

between major transcriptional shifts, and their presence within our pseudo-timeline validates our computational approach.

Plotting gene expression values along the pseudo-time axis provides a detailed insight into the dynamic expression patterns of these early transcribed genes. The published minor ZGA gene dataset[4] utilized in this analysis covered a tight developmental time window between NC 7 and 9, providing a static picture of an approximately 30-min-long developmental window. In contrast, our analysis provides previously unprecedented resolution of the minor ZGA, showing a staggered onset of transcription for these genes (Figures S2A and S2B). Intriguingly, many of the 20 minor ZGA genes share a similar sharp, transient peak of expression within the 3-h time window, indicative of their roles in early developmental processes. The exceptions are *E(spl)m4-*

*BFM*, a member of the Notch signaling pathway; *sisA*, a gene involved in sex determination; and *CG6885*, a gene of unknown function.

To identify the start of the major ZGA in our timeline, we used the combined expression levels of 17 genes that reportedly increase ($\geq$5-fold) between NC 14A and NC 14B (Figures 2D and 2F). These genes show increased expression at the transition from cluster 4 to 5 in our data (Figure 2F). Although the published gene list was curated using embryos within a 15-min developmental time window, carefully staged according to time elapsed in interphase, nuclear elongation, and progression of cellularization,[3] our continuous analysis shows that some of these genes actually increase transcription at earlier time points (Figures S2C and S2D). Taken together, our results provide the

**Table 1. Number of embryos in each quartile of the pseudo-time by collection time**

| Pseudo-time quartiles (Q) | Number of embryos in collection time intervals | | |
|---|---|---|---|
| | 0–1 h | 1–2 h | 2–3 h |
| Q1 | 22 | 4 | 0 |
| Q2 | 2 | 17 | 7 |
| Q3 | 5 | 7 | 14 |
| Q4 | 10 | 6 | 10 |

most detailed picture of the onset and dynamics of expression of previously reported ZGA transcript levels during early development that has been reported to date.

To confirm the dynamic nature of expression patterns uncovered in our dataset, we compared the expression dynamics among a select group of genes that are known to be transcribed early (*screw*, *zerknullt*, *spitz*, *deadhead*, *stumps*, and *yolkless*). These genes were previously shown to exhibit dynamic expression during development in different datasets that relied on the visual assessment and manual separation of samples into specific developmental categories or stages.[2,3] We plotted the normalized expression levels for these genes, according to the stages disclosed in datasets published by Sandler and Stathopoulos[3] (Figure 2G) and Lott et al.[2] (Figure 2H) and according to our new computationally determined timeline (Figure 2I). Graphs revealed that the transcriptional changes uncovered by our pseudo-time order are in good agreement with the previously published data. Pseudo-time order, cluster number, and sample ID for each embryo are shown in Table S2. Together, the results show that our method provides a high-resolution, time-sensitive picture of transcriptional events during early embryonic development.

### Single-embryo RNA-seq reveals novel early transcribed genes

To determine if our method can identify novel early expressed genes, we compared expression of genes between embryos from cluster 1 and cluster 2 of the t-SNE map displayed in Figure 2A. Within our dataset, we found the differential upregulation of transcription for 66 genes (adjusted p [padj] <0.01, Log2FC > 1) in this time frame (Figure 3A; Table S3). Over-representation analysis (ORA) shows that these genes are involved in sex determination and developmental processes (Figure 3B).

To validate their early expression and determine the biological age at which these transcripts are activated, we compared our dataset with the most comprehensive dataset on early zygotic transcription published to date[4] and performed qPCR on a subset of genes from samples obtained from hand-staged fixed embryos spanning NC 6 to 11. The results from our single-embryo and qPCR analysis confirm the published evidence that *sc* is one of the earliest expressed genes at NC 7 (Figure S3A). Our results also corroborate the early expression of the additional 19 genes previously reported to be expressed during NC 7–9. However, our analysis identified many other genes that are upregulated between clusters 1 and 2, which were previously reported to be expressed at significantly later time points, including 31 genes at NC 9–10 and 15 genes at syncytial blasto-

derm. Importantly, qPCR results validated our single-embryo analysis approach and confirmed the observed earlier onset of expression for a randomly selected subset of genes (Figures S3B–S3E; *ato* and *CG13465* were previously reported at NC 9–10 and *halo* and *dpn* were previously reported during syncytial blastoderm). We next wanted to exclude the possibility that these findings were due to the contamination of our qPCR samples with older embryos. To this end, we measured the levels of two gene transcripts (*hrg*, *bnb*) identified to be expressed at a later time point in both our own temporal analysis (Figures 3D, S3F, and S3G, right panels) and in other published datasets.[2,4] Using this approach, we detected no increase in expression for either *hrg* or *bnb* in the NC 7 and NC 8 samples (Figures S3F and S3G, left panels), which therefore excludes the possibility of contamination of our NC 7 and 8 samples with older embryos. Further analysis revealed an additional upregulation of 37 genes between clusters 2 and 3, including *hrg* and *bnb* (Figure 3D), and 111 genes between clusters 3 and 4 (Figure 3G; Table S3) with an enrichment in pathways related to early developmental processes (Figures 3E and 3H). Our analysis identifies 214 genes that are significantly upregulated between clusters before the onset of the major ZGA. Taken together, these results show that our approach is able to identify the accurate onset of transcription of early-expressed genes with high sensitivity.

We next compared expression between clusters 4 and 5 to identify genes activated at the beginning of the major ZGA (Figure 3J; Table S3). We identified 153 significantly upregulated transcripts in the cluster 5 embryos (padj < 0.1, Log2FC > 1). ORA revealed developmental pathways involved in tissue development, sex differentiation, and signaling pathways (Figure 3K).

Previous studies have shown that a small subset of zygotically transcribed genes are dependent on the transcription factor Zelda (Zld),[23] while a majority of zygotically transcribed genes are Zld independent but enriched for the histone variant H2A.Z.[24] To explore Zld dependency and H2A.Z occupancy behavior over time in our dataset, we quantified the overlap between our up- and downregulated transcripts in Figures 3A, 3D, 3G, and 3J in terms of being Zld dependent or independent while being bound or unbound by H2A.Z (Zld dependent and Zld independent, which were divided into H2A.Z positive or negative).[24] Our analysis shows that the earliest minor ZGA genes are mostly Zld dependent (Figures 3C and 3F) and that the share of Zld-dependent genes decreases sharply with the onset of the major ZGA. In contrast, the share of Zld-independent genes, both H2A.Z positive and negative, increases with the onset of the major ZGA (Figures 3I and 3L). These observations are not true for downregulated transcripts, which showed a similar distribution between being Zld dependent and independent, which was similar to all analyzed transcripts (Figure S3H).

In this way, we have demonstrated that our single-embryo RNA-seq methodology and analysis are a highly sensitive approach for identifying the accurate onset of gene transcription. Further, our analysis is able to define important transcriptional events and identify signatures of gene regulation during early development.

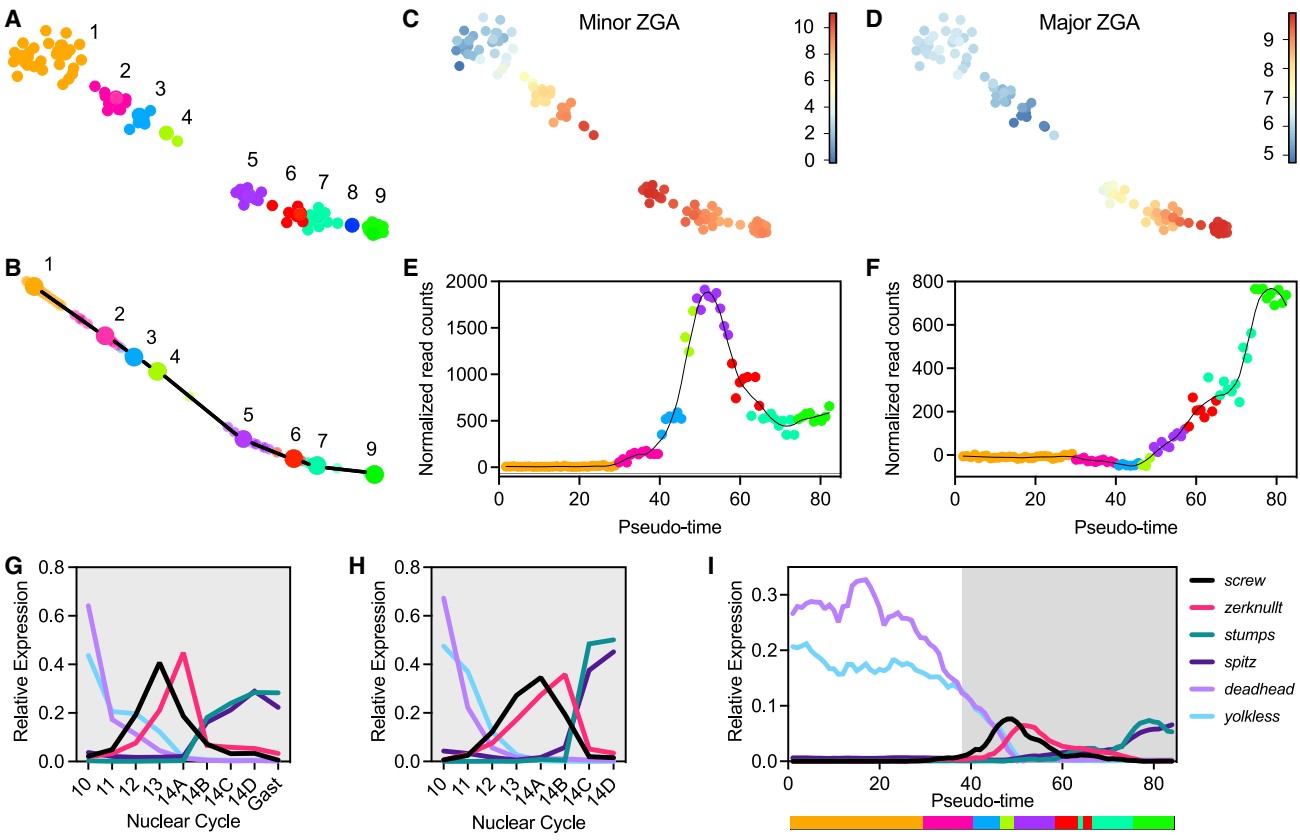

**Figure 2. The continuous sequence of the ZGA**

(A) t-SNE map visualization of embryos 10 min to 3 h old with k-medoid clustering indicated by different colors.

(B) Lineage analysis showing a single trajectory for all clusters (n > 1 embryos per cluster) leaving a total of eight clusters (n = 84).

(C and D) t-SNE map with coloring of individual dots according to the combined log2-transformed expression for 20 or 17 genes expressed during the minor (C) or major (D) wave of the ZGA (Table S1).

(E and F) Normalized read counts of minor (E) and major (F) ZGA genes for each embryo plotted along the pseudo-time order. The line represents the local regression of expression values on the ordered embryos.

(G–I) Relative expression of select genes from manually staged embryos reported by (G) Sandler and Stathopoulos,[3] (H) Lott et al.,[2] or (I) our computational age (pseudo-time). Gast, gastrulation. Gray background indicates the same developmental times included across datasets. For reference, the bar below the x axis in (I) indicates clusters according to their color. Normalized reads ordered by pseudo-time, and gene details can be found in Tables S7 and S8.xlsx.

## mRNA decay of maternally deposited transcripts

In addition to transcription, our dataset also reveals patterns of maternal RNA decay. While our method is unable to distinguish maternal mRNA from zygotic mRNA, it can be used to study maternal mRNA decay before the zygote takes complete control of its own transcription. In order to identify maternally degraded mRNAs only, we compared cluster 1 (youngest embryos) with cluster 5 (onset of major ZGA) and selected only maternally deposited transcripts for analysis. Maternally deposited transcripts were defined as those with an averaged normalized read count >1 on the first 10 samples in our pseudo-time. Using this method, a total of 2,621 significantly degraded transcripts were identified (padj < 0.01, Log2FC < −1). Ninety-two percent of these significantly degraded transcripts had also been shown to be degraded in a previously published dataset (classes II, III, IV, and V) (Figure 4A);[13] only 35, 33, or 13 genes were within the Thomsen stable (class I), purely zygotic group of transcripts, or preloaded and transcribed, respectively.

ORA of all 2,621 degraded transcripts revealed mainly pathways related to metabolism (Figure S4). This result likely reflects the elimination of transcripts important during oogenesis but that are no longer needed for development. While patterns of transcript abundance differed before and after cluster 5, there is a clear inflection point at around value 50 of our pseudo-timeline. This time point coincides with the onset of the major ZGA.

Maternal transcripts are deposited in oocytes at very different levels. To determine if degradation rates in the zygote are related to the initial quantity of deposited transcripts, we divided the significantly downregulated genes into four quartiles by their level of transcript abundance in cluster 1. We then determined the total number of normalized reads for each quartile in each individual embryo. Mean read counts plotted in Figure 4B show the progressive nature of the maternal mRNA decay up to cluster 5. Plotting the ratio of cluster 5 to cluster 1 for the different quartiles shows that the rate of decay is directly proportional to initial

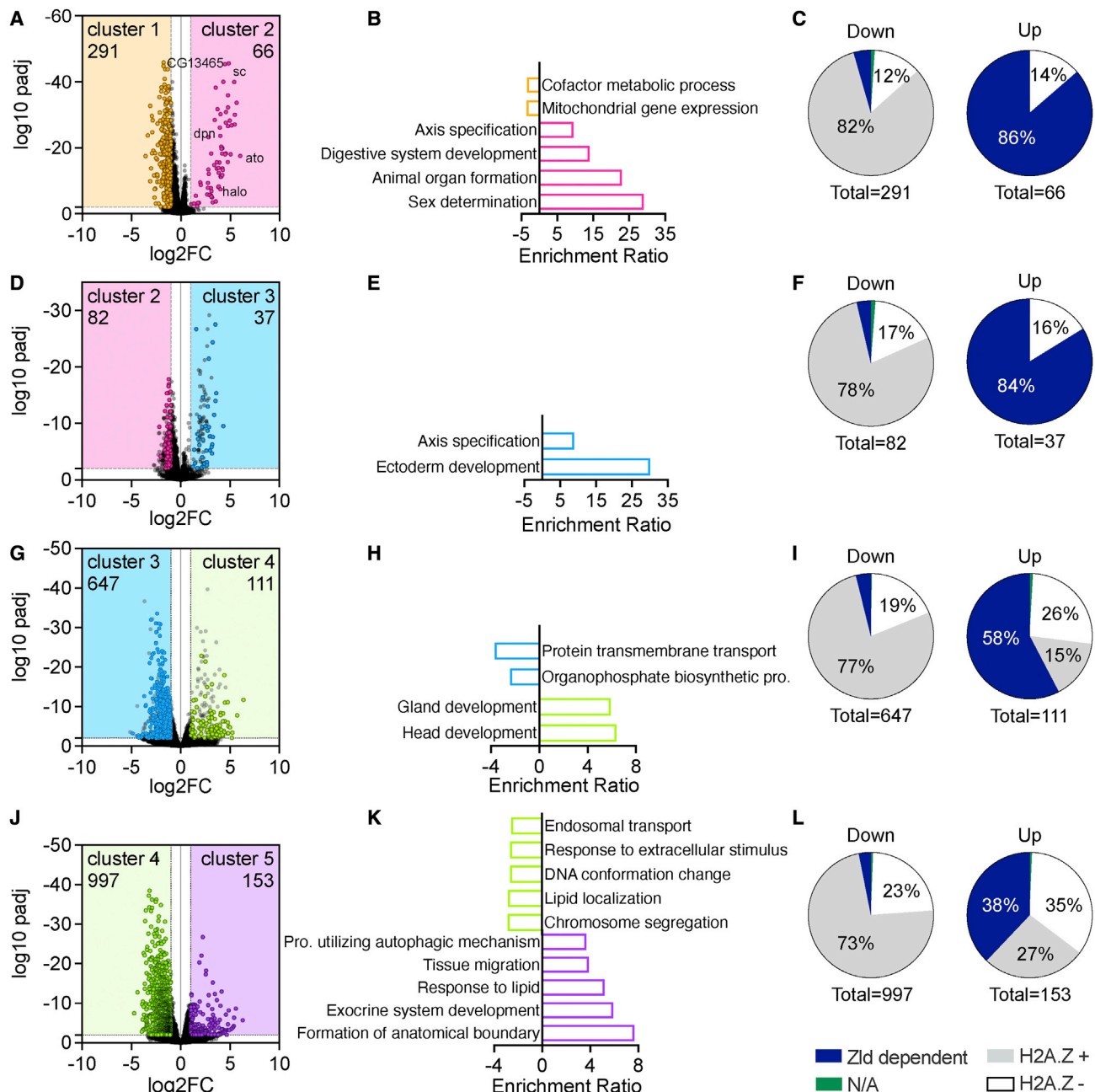

**Figure 3. Differentially expressed genes, their related pathways, and Zld or H2A.Z enrichment at TSS during the minor ZGA**

(A, D, G, and J) Volcano plots with significantly expressed genes (padj < 0.01, Log2FC < −1 or >1) by comparing (A) cluster 1 versus 2, (D) cluster 2 versus 3, (G) cluster 3 versus 4, and (J) cluster 4 versus 5. (A, D, G) The significantly changed unique transcripts that were not identified in previous cluster comparisons are represented by colored dots, and their numbers are indicated in each volcano plot. (J) Colored dots and numbers indicate all significantly expressed genes.

(B, E, H, and K) Significantly enriched pathways (false discovery rate [FDR] <0.05) by ORA on significantly expressed genes by comparing (B) cluster 1 versus 2, (E) cluster 2 versus 3, (H) cluster 3 versus 4, and (K) cluster 4 versus 5. Pro., process.

(C, F, I, and L) Zld and H2A.Z enrichment at TSS (transcriptional start sites) of differentially expressed genes between (C) cluster 1 versus 2, (F) cluster 2 versus 3, (I) cluster 3 versus 4, or (L) cluster 4 versus 5. Zld data from Blythe and Wieschaus,[23] and H2A.Z enrichment from Ibarra-Morales et al.[24] Genes not matching between datasets are shown as N/A.

mRNA abundance (Figure 4C), meaning that transcripts of low and high abundance are degraded at the same rate. This also provides evidence that our single-embryo approach does not introduce any bias toward lowly expressed genes, which would be affected most significantly by an overall increase in RNA content due to ZGA.

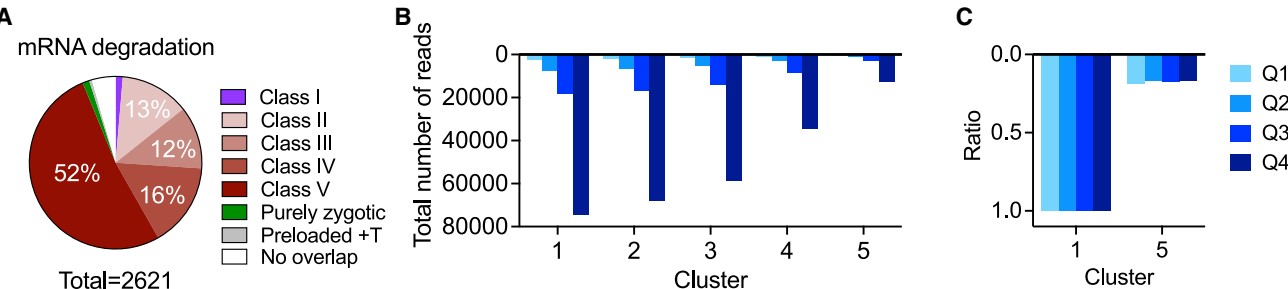

**Figure 4. The continuous mRNA decay of maternally deposited transcripts**
(A) Pie chart representing the different degradation classes previously reported by Thomsen et al., 2010,[13] for our maternally deposited transcripts significantly decreased (padj < 0.01, Log2FC < −1) between cluster 1 versus 5 + T = transcribed.
(B) Mean read counts of all significantly decreased transcripts (n = 2,621) by expression level group (Q, quartile) in each cluster. Q1, lowest 25%; Q4, highest 25%.
(C) Same data as in (B) showing the ratio of cluster 5 to cluster 1 by expression level.

Overall, we showed that maternal mRNA decay before the major ZGA is a progressive process. Degradation of maternal mRNA is proportional to transcript levels, suggesting that mRNA abundance is not related to degradation rate.

### X/Y chromosome genotyping uncovers transcriptional dynamics of primary sex determination

Our prior analysis of the earliest transcribed genes indicates sex determination as the most enriched pathway (Figure 3B). The current model for primary sex determination is based on the tightly controlled sex-specific expression levels of genes such as *Sex lethal* (*Sxl*) and *male-specific lethal* (*msl-2*)[25] during early development. This made us wonder if there are additional detectable differences in transcription between male and female embryos during our early developmental time window. To define the sex of each individual embryo, we isolated DNA from the organic phase after TRIzol extraction of RNA and performed qPCR using primers specific for the X and Y chromosomes. Due to low DNA content of younger embryos, we only get consistent PCR results after embryo number 23 in our pseudo-time analysis. Based on these results, we categorized all embryos (after pseudo-time position 23) according to sex (Table S2). To determine the differential expression of genes between male and female embryos, we used splineTimeR (see STAR Methods), which is particularly designed for identification of expression changes in longitudinal data. Our analysis identified 120 transcripts that were differentially expressed between male and female embryos (padj < 0.01) (Figure 5A). Although a large number of the differentially regulated genes are located on the X chromosome (44%), more than half the genes are located on autosomes and rDNA (56%). Several known regulators of primary sex determination, such as *Sxl*, *sc*, *sisA*, and *msl-2*, were also identified as significantly expressed in our analysis (Table S4). Indeed, ORA shows that sex differentiation is the most enriched pathway (Figure 5B). We selected and plotted known regulators of sex determination using our pseudo-timescale (Figures 5C and 5D). This analysis shows that differential transcription of *sc* and *sisA* (Figure 5D) precedes the expression of *Sxl*. This agrees with the role of *sc* and *sisA* as activators of *Sxl* expression in females (Figure 5C). Additionally, we identify other differentially expressed transcripts that precede *Sxl* transcription, such as *CG14427*. Ex-

amples for male-specific expression are plotted in Figure 5E. These data show that the start of differential expression of *stonewall* (*stwl*, chromosome 3L) matches the start of differential expression of *Sxl* and *msl-2* and that a pre-rRNA gene (*pre-rRNA:CR45847*) is expressed only in males shortly after *Sxl* and *msl-2*.

Another important process linked to primary sex determination is dosage compensation, which ensures equal expression of X-linked genes in males and females. In *Drosophila*, this is accomplished by the 2-fold upregulation of the X chromosome in males. It has previously been reported that X-lined genes are upregulated in males as early as NC 14,[2] but functional studies suggest the likely onset of canonical dosage compensation at NC 15.[12] To assess the onset of dosage compensation in our dataset, we excluded all maternally deposited transcripts and determined the total number of normalized reads for the remaining genes on the X chromosome and autosomes for each individual embryo. Average read counts of male and female embryos within each cluster are plotted in Figures 5G and 5H. From cluster 4 to 7, we observed a 1.5× higher expression of X-linked genes in female compared with male embryos, but no difference in autosomal reads. This difference was reduced to 1.1× in cluster 9 (gastrulation onset) (Figure 5G), probably due to the start of canonical dosage compensation. Two components of the dosage compensation complex have been shown to have male-biased transcription, *msl-2* and *long non-coding RNA on the X 1* (*lncRNA:roX1*). *msl-2* is expressed at higher levels in males almost from the moment it begins to be transcribed (Figure 5C) and well before we detect dosage compensation. *lncRNA:roX1* is expressed at higher levels in female embryos at first (Figure 5F), which can be explained by its localization on the X chromosome; levels only start to be higher in males once we see evidence for dosage compensation (cluster 8).

To further investigate how transcript levels are influenced by their chromosomal localization, we plotted early transcribed genes from our prior analysis according to sex and chromosome location (Figures 2A and 2D). Analysis shows that early transcribed genes from the X chromosome, but not autosomes, tend to have higher levels in females compared with males (Data S1). The time point at which expression levels in females

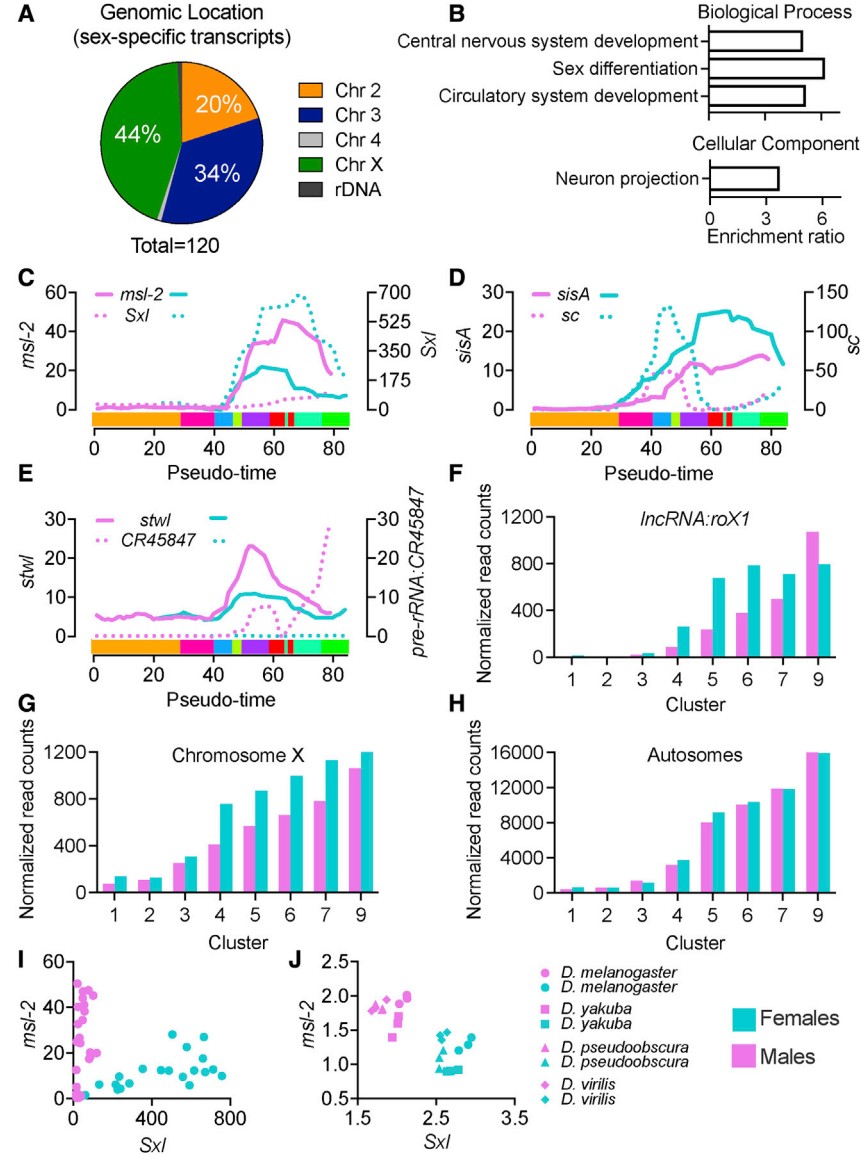

**Figure 5. Sex-specific transcription and dosage compensation in the ~3-h embryo**

(A) Distribution of differentially expressed genes (padj < 0.01) between male and female using splineTimeR according to their genomic locations.

(B) Significantly enriched pathways (FDR < 0.05) of differentially expressed genes by ORA.

(C–E) Smoothed normalized reads of selected transcripts. The colored bar along the x axis shows clusters 1–9 from left to right, each in a different color for reference.

(F–H) (F) Average normalized reads for *lncRNA:roX1* or all zygotic transcripts (not maternally deposited) from male and female embryos within each cluster, (G) for X-linked genes, or (H) autosomal genes.

(I) *msl-2* and *Sxl* normalized read counts of all male and female embryos in our data.

(J) *msl-2* and *Sxl* fragments per kilobase of transcript per million fragments mapped (FPKM) of males and females from different *Drosophila* species reported in Paris et al.[26] Metadata for sex-specific analysis can be found in Table S9.

the onset of *Sxl* and *msl-2* transcription, around NC 12 and NC 14D, respectively (Data S1D).

Overall, our analysis detects sex-specific transcription as early as the minor ZGA. Capitalizing on this differential gene expression, a simple strategy to sex embryos has been developed.

## DISCUSSION

Studying early development is challenging due to the rapid progression of biological processes and the limited amount of material available. To overcome these limitations, we developed a single-embryo RNA-seq and analysis approach, using the transcriptome as a measure of developmental progress (pseudo-time) to determine the biological age of the em-

are higher than in males varies between genes, with some differences detected as soon as transcription of a gene starts (e.g., *ac*, *acheate*) and others occurring later in transcription (e.g., *run*, *runt*) (Data S1).

Beyond the biological relevance of sex-specific transcription, we asked whether sex-specific gene expression could serve as a tool to determine the sex of individual embryos. This approach would eliminate the need for utilizing the standard time-consuming genotyping approach. To this end, we plotted *Sxl* and *msl-2* transcript levels and observed a clear separation of embryos according to their sex (Figure 5I). We further validated this approach by applying it to a published single-embryo sequencing dataset,[26] confirming that using *Sxl* and *msl-2* expression is sufficient to determine the sex in embryos of different *Drosophila* species (Figure 5J). Of note, this approach only works in embryos after

bryo. The high sensitivity of our method allows us to provide an accurate assessment of zygotic transcription and uncover the dynamic patterns for hundreds of genes. Our single-embryo approach also enables us to determine sex-specific differences in transcript abundance. Utilizing these sex differences, we developed a new strategy to determine the sex of each embryo, without the need for genotyping. Together, we established an operationally simple method to document gene expression changes at unprecedented temporal resolution and provide a continuous assessment of transcriptional processes during early development.

### An operationally simple, single-embryo sequencing method

Previous studies investigating zygotic transcription relied on precise collection time windows and/or elaborate manual staging of

the embryos under a microscope. *Drosophila* females, however, are known to lay unfertilized eggs and withhold embryos.[19] Human error in staging of embryos and irregular laying patterns of females can lead to the inclusion of mis-staged embryos in the analysis. Recent studies have highlighted the advantages of single-embryo sequencing approaches over working with pooled samples,[26,27] specifically the ability to detect and exclude older embryos from the analysis, therefore providing more accurate data.

In this work, we present an optimized single-cell sequencing protocol for use with *Drosophila* embryos along with a single-cell bioinformatic pipeline for their analysis. We assign a computationally derived age to each embryo, based upon their transcriptome, thereby circumventing the need for elaborate and error-prone staging procedures. Indeed, we show that our computationally derived pseudo-timeline reflects the biological age of the embryos, by comparison with previously established datasets. In addition, our protocol reduces reagent and sequencing costs due to the low-volume nature of the experiments and the inclusion of UMIs.[21] Further, this method requires no special instrumentation beyond a micro-pipetting device, and the analysis utilizes established tools.[22] Taken together, we show that sequencing 192 individual embryos provides a sufficient dataset to carry out detailed analysis of gene expression patterns over the first 3 h of development. Typically, it cost us around US$36 to generate the library and sequence a single embryo at 6 M reads, making it an affordable alternative to sequencing bulk samples of manually staged embryos. Together, these advantages make the method reported here the most accessible methodology developed to date, opening up this type of research to almost any *Drosophila* lab. This single-embryo sequencing approach will, ultimately, lead to improved reproducibility of developmental studies between experiments and laboratories.

### An accurate characterization of early transcriptional events

Our pseudo-time approach allows us to identify the exact onset of transcription even for lowly expressed genes and reveals that previously reported, as well as many novel, transcripts are expressed as early as NC 7. One example is the early expression of *halo*, a cofactor for the molecular motor, kinesin, and a regulator of lipid droplet movement, which was previously reported to be actively transcribed during syncytial blastoderm (after NC 11),[4] but was identified as one of the earliest transcribed genes in our dataset. Our analysis also reveals the dynamic nature of transcriptional events and provides information about expression for thousands of genes at a temporal resolution unchallenged by other methods. Recently, a single-cell dataset was published covering all of embryogenesis and providing insights into cell-type-specific transcriptional changes during development.[28] While this dataset provides an extremely detailed insight into *Drosophila* development, it only detected a median of 399 UMIs and 274 genes per cell, likely only covering very highly expressed genes. In contrast, we detected a median of over 600,000 UMIs and identified over 7,200 genes per embryo, leading to a total 9,777 identified genes across the whole dataset. Thus, our dataset gives a much more complete picture of transcriptional changes during early development.

### Sex-specific gene expression

Our single-embryo method also distinguishes between male and female embryos, allowing for investigations into sex-specific transcriptional effects. X-signal elements (XSEs) have been shown to control the early sex-specific expression of *Sxl* and to drive primary sex determination. We show differential expression of *sc* and *sisA*, two strong XSEs, from the very first moment of ZGA (NC 7). Of note, CG14427, an X-linked gene with unknown function, is also differentially expressed between males and females starting at NC 7, making it a potential candidate as a novel XSE. Our data also allow for further insights into primary sex determination. An early expressed XSE, *run*, was previously reported to undergo a non-canonical form of dosage compensation and expression that was proposed to be under the direct control of *Sxl*.[29,30] Our results show that *run* is one of the earliest transcribed genes, preceding expression of *Sxl*. Our data also show that expression of *run* peaks after *Sxl* peak expression. As such, our data support a role of *run* as a regulator of *Sxl*[31] rather than *Sxl* controlling expression of *run*.[29,30] Surprisingly, we also identified several autosomal encoded genes as differentially expressed between males and females. While differential expression of X chromosome genes in females can be explained by their different dosage (2X in females versus 1X in males), this is not the case for autosomal genes, which are present at equal dosage in both males and females. These results suggest additional players in primary sex determination. Further studies will be needed to confirm these results and investigate the underlying mechanisms. Importantly, our newly developed strategy to determine the sex of single embryos by using the expression of known regulators of primary sex determination (*Sxl* and *msl-2*) eliminates the need for elaborate genotyping procedures in future sequencing datasets.

### Limitations of the study

While our method detects more transcripts (9,777) and achieves a higher sequencing depth (600,000) than single-cell methods, it may still be more difficult to detect lowly expressed genes compared with bulk RNA-seq datasets. Further, our dataset lacks cell-type-specific information. However, this is likely of only minor relevance during early development, as it is known that all nuclei share the same cytoplasm until cellularization, and the previously reported single-cell sequencing dataset only identified three different types of cells (anlage *in statu nascendi*, aminoesra anlage, ectoderm anlage) during the first 4 h of development.[28] That said, we acknowledge that cell-specific expression patterns become more important at later stages in development, and, while our method does not allow for this kind of analysis, our approach remains much more accessible than single-cell sequencing of thousands of individual cells at different developmental time points. Our method also does not reveal information about the spatial distribution of transcripts. Transcript gradients, however, have been shown to be of particular importance during early development. Once changes in gene expression have been determined using our methods, these patterns could be further investigated using *in situ* hybridization to assess whether the spatial distribution of a particular transcript is altered.

Taken together, we believe our method is the most accessible, high-throughput, transcriptomic technology published to date to

study early gene expression in *Drosophila*. We suggest that our methodology provides the optimal tool to investigate the transcriptional consequences of mutations in developmental genes, providing gene expression data at a depth and temporal resolution that was previously inaccessible.

## STAR★METHODS

Detailed methods are provided in the online version of this paper and include the following:

- KEY RESOURCES TABLE
- RESOURCE AVAILABILITY
  - Lead contact
  - Materials availability
  - Data and code availability
- EXPERIMENTAL MODEL AND SUBJECT DETAILS
  - Organisms
- METHOD DETAILS
  - Embryo collection
  - RNA isolation from single embryos
  - Library preparation and RNA-seq
  - RNA-seq data analysis and functional enrichment
  - Fixation, staining, and staging of embryos for qPCR analysis
  - Reverse transcription on pooled RNA samples
  - DNA extraction on single embryos for sex determination
  - qPCR and qPCR data analysis
  - Zelda-dependent/independent and H2A.Z-positive/negative genes
  - Sex-specific transcription

## SUPPLEMENTAL INFORMATION

## ACKNOWLEDGMENTS

We thank Carolyn Anderson for critical reading and editorial feedback on the manuscript. We are grateful to all members of the Lempradl lab for helpful discussions. We thank the Van Andel Institute Genomics Core, especially Marc Wegener and Marie Adams, for their assistance with RNA-seq. We thank the VAI Bioinformatics and Biostatistics Core for supporting the analyses. We thank J. Andrew Pospisilik for supporting the project.

## AUTHOR CONTRIBUTIONS

J.E.P.-M. and A.L. designed and directed the study. J.E.P.-M., J.W., and L.E. performed experiments. J.E.P.-M., K.L., and A.L. analyzed the data. J.E.P.-M. and A.L. wrote the manuscript.

## DECLARATION OF INTERESTS

The authors declare no competing interests.

## INCLUSION AND DIVERSITY

One or more of the authors of this paper self-identifies as a gender minority in their field of research. One or more of the authors of this paper self-identifies as a member of the LGBTQIA+ community.

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

## STAR★METHODS

### KEY RESOURCES TABLE

| REAGENT or RESOURCE | SOURCE | IDENTIFIER |
|---|---|---|
| **Chemicals, peptides, and recombinant proteins** | | |
| Fly food M | LabExpress | Cat#7002 |
| TRIzol[TM] Reagent | Thermo Fisher Scientific | Cat#15596026 |
| GlycoBlue[TM] Coprecipitant | Thermo Fisher Scientific | Cat#AM9516 |
| ERCC RNA Spike-In Mix | Thermo Fisher Scientific | Cat#4456740 |
| VaporLock | Qiagen | Cat#981611 |
| RNaseOUT[TM] Recombinant RNase Inhibitor | Thermo Fisher Scientific | Cat#10777019 |
| Second Strand Buffer | Thermo Fisher Scientific | Cat#10812014 |
| DNA Polymerase I | Thermo Fisher Scientific | Cat#18010025 |
| *E. coli* DNA Ligase | Thermo Fisher Scientific | Cat#18052019 |
| Ambion[TM] RNase H, from *E. coli*, 10 U/μL | Thermo Fisher Scientific | Cat#AM2293 |
| AMPure XP reagent | Beckman Coulter | Cat#A63881 |
| RNAClean XP | Beckman Coulter | Cat#A63987 |
| ExoSAP-IT PCR Product Cleanup Reagent | Thermo Fisher Scientific | Cat#78200.200.UL |
| Phusion® High-Fidelity PCR Master Mix with HF Buffer | New England Biolabs | Cat#M0531S |
| **Critical commercial assays** | | |
| SuperScript[TM] II Reverse Transcriptase | Thermo Fisher Scientific | Cat#18064014 |
| MEGAscript[TM] T7 Transcription Kit | Thermo Fisher Scientific | Cat#AM1333 |
| NEBNext® Magnesium RNA Fragmentation Module | New England Biolabs | Cat#E6150S |
| **Deposited data** | | |
| Zelda-dependent/independent and H2A.Z-positive/negative genes data | Ibarra-Morales et al.[24] | Data S1 |
| Single-embryo RNA-seq raw data | This paper | GEO: GSE214118 |
| Single-embryo RNA-seq raw reads | This paper | GEO: GSE214118 |
| **Experimental models: Organisms/strains** | | |
| *D. melanogaster*: DGRP-737 | Bloomington Drosophila Stock Center | BDSC:83729 |
| **Oligonucleotides** | | |
| CEL-Seq2 oligoT: GCCGGTAATACGACTCACTATAGGGAG TTCTACAGTCCGACGATCNNNNNNN[6 base barcode]TTTTT TTTTTTTTTTTTTTTTTTTTV | Hashimshony et al.[20] | Table S2 |
| CEL-Seq2 library RT primer: GCCTTGGCACCCGAGAATTC CANNNNNN | Hashimshony et al.[20] | Table S2 |
| RP1 primer: AATGATACGGCGACCACCGAGATCTACACGT TCAGAGTTCTACAGTCCGA | Sagar et al.[21] | Table 2 |
| RPI1 primer: CAAGCAGAAGACGGCATACGAGATCGTGAT GTGACTGGAGTTCCTTGGCACCCGAGAATTCCA | Sagar et al.[21] | Table 2 |
| RPI2 primer: CAAGCAGAAGACGGCATACGAGATACATCG GTGACTGGAGTTCCTTGGCACCCGAGAATTCCA | Sagar et al.[21] | Table 2 |
| qPCR primers | This paper | Table S6 |
| **Software and algorithms** | | |
| fastqc (v0.11.9) | Andrew, S.[32] | https://www.bioinformatics.babraham.ac.uk/projects/fastqc/ |
| STAR (v2.7.8a) | Dobin et al.[33] | https://github.com/alexdobin/STAR |
| RaceID3/StemID2 | Herman et al.[22] | https://github.com/dgrun/RaceID3_StemID2_package |
| Prism (v9.4.1) | GraphPad Software | https://www.graphpad.com |

## RESOURCE AVAILABILITY

### Lead contact

Further information and requests should be directed to and will be fulfilled by the lead contact, Adelheid Lempradl (heidi.lempradl@vai.org).

### Materials availability

This study did not generate new unique reagents.

### Data and code availability

- Single-embryo RNA-seq data from this study has been deposited at NCBI Gene Expression Omnibus (GEO). Accession numbers are listed in the key resources table. Normalized reads, gene details, and metadata for sex-specific analysis can be found in Tables S7, S8, and S9.
- All original code is available in this paper's supplemental information.
- Any additional information required to reanalyze the data reported in this paper is available from the lead contact upon request.

## EXPERIMENTAL MODEL AND SUBJECT DETAILS

### Organisms

Drosophila genetic reference panel (DGRP)-737 line from Bloomington Stock Center (#83729) was kept in incubators at 25°C with 60% humidity and a 12-h light-dark cycle. All flies were raised at constant densities on standardized cornmeal food (Bloomington recipe), Fly food M (LabExpress, Michigan, USA), and transferred into cages 1-2 days after eclosion.

## METHOD DETAILS

### Embryo collection

8-9 day old flies were put in embryo collection cages (Genesee Scientific, Cat #: 59-100). Food plates were changed and discarded twice before embryo collection started on. DGRP-737 line showed minimal egg laying (n = 0–2) in the first 30 min after changing the plates (data not shown), therefore, plates were changed every 90 min and processed immediately (0-1 h embryos) or incubated 1 or 2 more hours at 25°C for 1-2 h or 2-3 h timepoints respectively. Embryos were transferred into a pluriStrainer 150 μM cell strainer (pluriSelect, USA) and washed with tap water. Embryos were dechorionated by incubation in 3% sodium hypochlorite (PURE BRIGHT bleach, KIK international LLC) for 4 min, washed in 120 mM NaCl (Sigma-Aldrich, USA), 0.03% Triton X-100 (Thermo Fisher Scientific, USA) solution, and finally washed in ultrapure water (PURELAB Ultra, ELGA). For RNA-seq, single embryos were transferred into 2 mL pre-labelled (on side and top) screw-cap microtubes containing 0.2 g lysing matrix D beads (MP Biomedicals, USA) using a 20/0 liner brush (Royal & Langnickel, USA), snap-frozen on dry ice, and stored at −80°C. Embryos for qPCR experiments were fixed immediately.

### RNA isolation from single embryos

500 μL TRIzol Reagent (Thermo Fisher Scientific, USA) and 50 μL Gibco 1x PBS (phosphate buffered saline) at pH 7.4 were added to microtubes with frozen single embryos. Samples were homogenized by bead-beating with 0.2 g lysing matrix D beads (MP Biomedicals, USA) at 6 m/s for 30 s using a FastPrep-24 instrument (MP Biomedicals, USA). RNA was then isolated following a miniaturized version of the manufacturer's instructions. Briefly, 100 μL chloroform (Sigma-Aldrich, USA) was added, samples mixed by vortex, incubated 2 min at room temperature (RT), and centrifuged for 15 min at 12,000 × g at 4°C. The aqueous (upper) phase was transferred to a new pre-labelled 1.5 mL microtube, keep samples on ice. At this step, the organic phase was stored at −80°C for later DNA extraction. RNA was precipitated by adding 250 μL ice-cold isopropanol (Sigma-Aldrich, USA) and 2 μL GlycoBlue (Thermo Fisher Scientific, USA) as coprecipitant. If processing more than 14 samples, it is recommended to split samples into two groups after this step. Samples were mixed by hand, incubated for 10 min at RT, and centrifuged for 10 min at 12,000 × g at 4°C. RNA pellets were washed with 1 mL 75% (v/v) ethanol (Pharmco, USA), dried, and stored at −80°C until further use.

### Library preparation and RNA-seq

RNA-seq was carried out following a miniaturized version of the sensitive highly-multiplexed single-cell RNA-seq (CEL-Seq2) protocol[20,22] with modifications.

Dried RNA was resuspended in 8 μL nuclease-free water (Thermo Fisher Scientific, USA) and a nanoliter-scale liquid handler was used to dispensed 120 nL of RNA sample into a 384-well plate holding 240 nL of primer-mix, ERCC RNA Spike-In Mix (Thermo Fisher Scientific, USA), dNTPs, CEL-Seq2 oligoT with different barcodes and unique molecular identifiers (UMI) and 1200 nL VaporLock (Qiagen, USA) to avoid evaporation. All pipetting steps referring to nL were carried out using a nano-scale automated liquid handler, and we have successfully used the i.DOT (CELLINK) and the mosquito LV (sptlabtech).

RNA was converted to cDNA by two reactions. First-strand synthesis was carried out by addition of 160 nL of reaction mix (0.08 μL 5X first-strand buffer, 0.04 μL 100 mM DTT, 0.02 μL SuperScript II and 0.02 μL RNaseOUT (Thermo Fisher Scientific, USA)), incubation for 1 h at 42°C, incubation for 10 min at 70°C, and a final incubation at 4°C for a minimum of 5 min. Second-strand synthesis was carried out by addition of 2200 nL reaction mix (1.62 μL nuclease-free water, 0.0525 μL dNTP mix (10 mM ea), 0.53 μL s strand buffer, 0.075 μL DNA Polymerase I, 0.02 μL *E. coli* DNA ligase, and 0.02 μL Ambion RNase H, from *E. coli*, 10 U/μL (Thermo Fisher Scientific, USA)) and incubation for 2 h at 16°C.

cDNA from 96 single embryos was pooled together and cleaned up with 0.8 μL AMPure XP reagent (Beckman Coulter, USA) per 1 μL of sample, recovering a total of 13 μL cDNA. *In vitro* transcription was performed in a 32 μL final volume reaction (13 μL cDNA and T7 enzyme mix, 10X T7 reaction buffer, ATP, GTP, UTP, and CTP 3.2 μL/each) using MEGAscript T7 Transcription Kit (Thermo Fisher Scientific, USA). After incubation for 16 h at 37°C, the amplified RNA (aRNA) was recovered and cleaned up by adding 12 μL ExoSAP-IT PCR Product Cleanup Reagent (Thermo Fisher Scientific, USA) and incubation at at 37°C for 15 min.

The resulting aRNA was fragmented for exactly 3 min at 94°C using NEBNext Magnesium RNA Fragmentation Module (5 μL fragmentation buffer plus 5 μL stop solution per reaction, NEB, USA). aRNA was cleaned up with 0.8 μL RNAClean XP (Beckman Coulter, USA) per 1 μL sample, recovering a total of 21 μL aRNA. 1 μL was used to check the size distribution on Agilent 2100 Bioanalyzer and the remaining 20 μL were mixed with 2 μL dNTP mix (10 mM ea) and 4 μL 10 μM CEL-Seq2 library RT primer. The mix was incubated for 5 min at 65°C and immediately chilled and kept on ice before adding 8 μL 5X first-strand buffer, 4 μL 100 mM DTT, 2 μL SuperScript II and 2 μL RNaseOUT. The 42 μL reaction mix was incubated for 10 min at 25°C and subsequent 1 h at 42°C to achieve first-strand synthesis.

cDNA was diluted 10-fold using nuclease-free water (v/v) before an 11-cycle PCR amplification in a 50 μL final volume reaction (10 μL cDNA, 11 μL nuclease-free water, 2 μL 10 μM RP1 primer, 2 μL RPI-1 or 2 (IDT, USA) and 25 μL 2X Phusion High-Fidelity PCR Master Mix with HF Buffer (NEB, USA)). Each amplification cycle included denaturation for 10 s at 98°C, annealing for 30 s at 60°C and extension for 30 s at 72°C. Library PCR included a pre-incubation for 30 s at 98°C and a final incubation for 10 min at 72°C. Libraries were cleaned up twice with 0.8 μL AMPure XP reagent per 1 μL sample, recovering a total of 11 μL sample.

Paired-end sequencing (150 bp) was performed using the NovaSeq 6000 instrument (Illumina) by the Genomics Core at Van Andel Institute. Sequencing depth in each single embryo was between 6.4 and 6.8 M reads that passed quality control, with 96% of the sequences with a quality score ≥30 (FastQC version 0.11.9).[32] Summary statistics of the RNAseq can be found in Table S5.

### RNA-seq data analysis and functional enrichment
RNA-seq read counts were demultiplexed, mapped to the Berkeley Drosophila Genome Project assembly release 6.28 (Ensembl release 100) reference genome,[34] UMI-deduplicated, and counted using STAR 2.7.8a (mode STARsolo).[33] Gene symbols were updated using release FB2022_04. Samples with a total transcript read count <250,000 or transcripts with <3 read counts in < 5 samples were filtered out from the analysis. Read count normalization, computation of a distance matrix, sample clustering, transcriptome entropy calculation generation of a lineage tree, and pseudo-temporal order of samples was carried out using R packages RaceID version 0.2.6 and FateID version 0.2.2.[22] Raw expression values of unsupervised clusters were compared by the RaceID3 internal approach akin to DESeq2. Transcripts with a Benjamini-Hochberg adjusted p value (padj) < 0.01 and a log2 fold-change (Log2FC) <-1 or >1 were considered to be differentially expressed. The source code for this analysis can be found in Data S2, Table S6.

All functional enrichment analyses were carried out by over-representation analysis (ORA) using the WEB-based GEne SeT AnaLysis Toolkit (WebGestalt).[35] using redundancy reduction by affinity propagation. Only pathways with a false discovery rate (FDR) ≤ 0.05 are shown.

### Fixation, staining, and staging of embryos for qPCR analysis
Dechorionated embryos were transferred to a 1.5 mL microtube and mixed in 362.5 μL PBT (0.3% Triton X-100 in PBS), 12.5 μL 10x PBS, and 125 μL 16% formaldehyde, methanol-free (w/v) (Thermo Fisher Scientific, USA). Embryos in the 4% formaldehyde fixing solution (w/v) were shaken for 15 min at 200 rpm using a mini rotator/shaker (Thermo Scientific). Fixing solution was discarded, 500 μL heptane (Sigma-Aldrich, USA) and 500 μL methanol (Thermo Fisher Scientific, USA) were added, and samples were vigorously shaken by hand/vortex for 2 min. Heptane, methanol, and embryos in the interphase were removed and discarded. Samples were washed 3 times with methanol before resuspension in 1 mL PBT containing 1 μL Hoechst 33,342 (20 mM) (Thermo Fisher Scientific, USA). After a 10 min incubation at RT, 2 × 1 min and 1 × 10 min washes with 1 mL PTB were carried out to remove excess dye. Embryos were then staged using the ECLIPSE Ts2 microscope (Nikon) based on Foe et al.,[10] and reference images for nuclear cycle divisions from others.[36,37] Embryos in PBT were kept on ice during staging. Finally, PBT was removed, TRIzol was added to pooled embryos, and samples were stored at −80°C until RNA was isolated using a standard TRIzol RNA isolation protocol.

### Reverse transcription on pooled RNA samples
Dried RNA from staged embryos was resuspended in 9 μL nuclease-free water and 1 μL used for quantification by NanoDrop One/OneC spectrophotometer (Thermo Scientific). The remaining RNA (<3 μg) was treated with 2 U TURBO DNase (Thermo Fisher Scientific, USA) following manufacturer's instructions. RNA was then incubated at 70°C with 1.5 μg oligo(dT)$_{12-18}$ (Thermo Fisher Scientific, USA) and immediately chilled on ice. Reverse transcription was carried out using moloney-murine leukemia virus

(M-MLV) reverse transcriptase kit (Promega, USA). Reverse transcription was completed in a 30 μL final volume reaction containing 400 U M-MLV and 1 mM dNTP mix after serial incubations at 40°C for 60 min and 90°C for 10 min cDNA was chilled on ice and diluted to a concentration of 20 ng/μL (1 μg input RNA/50 μL).

### DNA extraction on single embryos for sex determination
DNA extraction was performed with a modified version of the manufacturer's instructions (TRIzol). The frozen organic phase of each embryo after RNA extraction was thawed at RT for 3 min and transferred to a fresh 1.5 mL microtube to remove beads from samples. 2 μL GlycoBlue coprecipitant were added, samples mixed by inverting tube 5 times, 150 μL 100% ethanol (Pharmco, USA) were added, and samples mixed again. After a 3 min incubation at RT samples were centrifuged 5 min at 7,000 g at 4°C and the phenol-ethanol supernatant discarded. DNA pellets were washed in 500 μL 0.1 M sodium citrate in 10% ethanol and incubated 30 min mixing every 10 min. Samples were centrifuged for 5 min at 7,000 g at 4°C and supernatant discarded. Wash with 0.1 M sodium citrate was repeated once, and pellets resuspended in 1 mL 75% ethanol. Then, 2 μL GlycoBlue coprecipitant were added, and samples incubated for 10 min mixing every 2-5 min. Samples were centrifuged 5 min at 7,000 g at 4°C, supernatant was discarded and pellet air dried before resuspension in 20 μL 8 mM NaOH in $H_2O$ (w/v). DNA samples were incubated at 80°C for 10 min mixing every 2 min by vortex, chilled immediately on ice for 5 min, and stored at 80°C.

### qPCR and qPCR data analysis
qPCR was carried out in a 20 μL final volume reaction using SsoAdvanced Universal SYBR Green Supermix (Bio-Rad, USA), bespoke forward/reverse primers (0.3 μM/each) (Table S6), and 2 μL DNA (1/10 embryo) or 160 ng/μL cDNA. Pre-incubation at 98°C for 3 min for DNA or 30 s for cDNA, 45 cycle amplification, and melting curve were performed using CFX96 touch real-time PCR detection system (Bio-Rad). Each amplification cycle included denaturation at 95°C for 10 s and a combined annealing/extension at 60°C for 30s. Specificity of qPCR reactions was assessed by the presence of a single peak in the melting curve, which was generated by acquiring fluorescence data every 0.5°C change in temperature from 65°C to 95°C. All qPCR reactions were performed in duplicates. DNA samples that amplified the X and Y chromosome in both duplicates at similar cycle threshold values were categorized as males. DNA samples that amplified the X but not the Y chromosome were categorized as females. For cDNA samples, mRNA expression in each duplicate was calculated using the cycle threshold values by the standard curve method.[38] The expression of the gene of interest was then divided by the geometric means of *CG6707* (FBgn0036058) and *Pgam5* (FBgn0023517), two transcripts with the lowest variability until around NC 14D in our RNA-seq data.

### Zelda-dependent/independent and H2A.Z-positive/negative genes
Completed using analyzed data available from Ibarra-Morales et al.[24] who identifed Zld-dependent, H2A1.Z-positive, and H2A1.Z negative genes. We used their classification to group differentially expressed genes (padj<0.01, Log2FC < −1 or >1) in our data, see Table S1.

### Sex-specific transcription
RNA-seq normalized read counts of each transcript were compared between male and female embryos using splineTimeR version 1.24.0.[39] Every embryo was considered a replicate in every cluster (timepoints). Transcripts with a Benjamini-Hochberg adjusted p value (padj) < 0.01 were considered significantly expressed. The source code for this analysis can be found in Data S2. Due to the split of the pseudo-time into male and females, normalized reads were smoothed by averaging 5 neighboring samples and a second order of the smoothing polynomial using Prism 9 version 9.4.1.

