## [Document S2. Transparent peer review record for Pérez-Mojica · Cell Genomics]

Continuous transcriptome analysis reveals novel patterns of early gene expression in *Drosophila* embryos

Author list

J. Eduardo Pérez-Mojica, Lennart Enders, Joseph Walsh, Kin H. Lau, and Adelheid Lempradl

Summary

Initial submission: Received : September 29th 2022

Scientific editor: Judith Nicholson

First round of review: Number of reviewers: 3
Revision invited : November 18th 2022
Revision received : December 12th 2022

Second round of review: Number of reviewers: 1
Accepted : 20th January 2023

Data freely available: Yes

Code freely available: Yes

This transparent peer review record is not systematically proofread, type-set, or edited. Special characters, formatting, and equations may fail to render properly. Standard procedural text within the editor's letters has been deleted for the sake of brevity, but all official correspondence specific to the manuscript has been preserved.

Referees' reports, first round of review

Reviewer 1

This study uses RNA-Seq on over a hundred single embryos, and computational methods similar to those used in single-cell experiments, to create a pseudo-timecourse of *Drosophila* embryonic development. This is a useful technique, modernizing previous approaches, and has produced an interesting and useful dataset.

My concerns are largely minor. I have two overarching concerns and a few smaller points:

1. The first matter, which comes up at several places throughout the manuscript, is the idea that hand staging of embryos is difficult and time-consuming. While somewhat time-consuming, staging embryos visually is not that difficult, indeed numerous undergraduate researchers I work with have been trained to do so consistently and reliably. The earlier experiments that were mentioned for incorporating erroneously staged embryos were performed at a time where proper replication was not possible due to the costs of sequencing, and so data from one or two embryos were used (for example, the cited Lott et al., 2011 had only one replicate for some timepoints as each embryo required an entire lane of Illumina sequencing at that time), now proper replication is much more affordable. While the method proposed in this manuscript is clearly the way that things are headed, it may be that the cost of sequencing close to two hundred embryos is prohibitive for some researchers, and hand staging a small number of replicates is more accessible. The authors might include some discussion about the number of samples needed for their method here, and the continued precipitous drop in sequencing costs could be used to motivate their method as the best one for the future.

2. In a couple of places in the text (detailed below), I have concerns related to how the method used to remove unfertilized eggs may affect their results. As reported (line 106), the authors use expression of previously reported early transcribed genes to remove unfertilized eggs from inclusion in their datasets. Presumably, then, these genes represent early zygotic transcription.

- In line 118, the authors claim that their pseudo-time starts at 10 minutes after

fertilization. As the method of removing unfertilized eggs also removes all timepoints before any zygotic transcription begins (when maternal transcripts entirely control development), it's difficult to understand how this can represent 10 minutes after fertilization, as it does not include any of the time before any zygotic transcription is present. Do the authors mean 10 minutes after the earliest zygotic transcription, instead? Please clarify.

- For the mRNA decay of maternally deposited transcripts section (line 235), again I have the same concern as above that the method used to remove unfertilized eggs necessarily includes some genes with zygotic transcription. While presumably only a small proportion of transcripts are zygotic at their first timepoint, if the point is to identify maternal genes that are not zygotically transcribed in order to examine the degradation of these transcripts, this is not a clean dataset to use for this purpose. I appreciate that the authors compare their data to an older dataset that is better designed for this purpose (Thomsen et al., 2010). I would suggest that the authors acknowledge this limitation in their dataset when describing this analysis.

Small points:

Line 250 - was there thought to be some relationship between the amount of an mRNA maternally deposited into the egg and the rate of degradation of that particular mRNA? I'm not sure mechanistically how that would work given what is known about degradation mechanisms. Perhaps it's interesting then to provide counter-evidence, as is done here.

Line 295 - Lott et al., 2011 shows that X chromosomal dosage compensation is not fully active by the end of NC 14, just that MSL-2 transcripts are present in males at a high level by what this study terms NC14C. This study posits a second, non-MSL mechanism that is active for some X-linked genes prior to MSL activation. Functional studies (such as Strukov et al., 2011, PLOS Biology) show likely onset of MSL-mediated dosage compensation around gastrulation/cycle 15.

Starting on line 396- discussion of runt. I think the discussion might be confusing the role of runt as an activator of Sxl with the potential role of runt as a Sxl target, which may be involved with the multiple roles of runt during this developmental

timeperiod (i.e. earlier as an XSE and later as a pair rule gene). Despite being an XSE runt is known to be relatively dosage insensitive, and recent evidence (Mahadeveraju, Jung, and Erickson, 2020, G3) showed that runt helps maintain expression from the establishment promoter of Sxl in female embryos. I would suggest that the authors revisit this section of the text as I think these findings (and more generally runt as a regulator of Sxl) help explain their observations.

Starting on line 542 - Fixation, staining, and staging of embryos. I couldn't find in the manuscript why embryos were fixed and stained? A number of the previous single-embryo methods that involve staging do not fix or stain the embryos. Please clarify for what datasets these methods were used, and for what purpose.

Reviewer 2

Due to the rapid progression and the limited amount of material, studying early embryo developmental events is challenging. This manuscript by Perez-Mojica and colleagues developed a scalable single-embryo RNA-sequencing and analysis approach to study zygotic gene expression during early embryogenesis in *Drosophila*. Using the transcriptome as a measure of development, the Authors are able to align >100 embryos in a linear trajectory to determine the developmental stages and uncover the exact onset of transcription and degradation. It is also useful for investigating sex-biased transcription from the beginning of zygotic transcription to gastrulation. This approach helps avoid timing bias due to the rate of egg fertilization and embryo withholding.

Overall, this manuscript is clearly written with proper citation and discussion of previous related studies. It will provide an important resource to reveal zygotic gene activation in a more precise manner. However, I have a few concerns that need be addressed before I can recommend it to be published.

Major points:

1. The novelty and all major findings of the current study are depending on the accurate organization of individual embryos along the pseudo-time. So, it is very important to ensure the trajectory is correct. The authors used different dimensionality reduction methods and showed these embryos form a linear organization. This is a good sign, but it doesn't guarantee all embryos are aligned in the same order. I suggest the Authors to use different trajectory analysis approaches, that have been developed and used for single cell RNAseq studies,

such as Monocle 3 and Velocity for RNA velocity, to validate and correct (if any wrong) for the pseudo-time analysis.

2. Most analyses are focused on validating previous findings, but not on newly identified factors. It will be nice to show new findings from current study. For example, how many new zygotic genes are identified? Which factors' activation timings should be updated? Showing such information will benefit the development community.

3. For figure 3 and 4, the Authors performed the comparison analysis in the cluster level. Within each cluster, is there significant embryo-embryo variation? Will this variation impact the cluster level comparison?

Minor points:

1. In *Drosophila*, XXY represents a female, XY represents a male. By sex-chromosome genotyping analysis, how to distinguish the gender? Did the authors remove embryos of XXY genotype?
2. It will be nice to show quality control data such as # UMI and # genes per embryo in Supplement.
3. For Fig 3, it is not clear how Zld dependent, H2A.Z positive or negative, are defined. Please add details in the Methods.
4. For Volcano plots in Fig 3, in the legend padj was used, but p-value was used in the plots.
5. In line 539: "Only results with a false discovery rate (FDR) ≤ 0.5 are shown." Should it be 0.05.

Reviewer 3

Perez-Mojica et al. present a new method for analyzing temporal dynamics of gene expression during *Drosophila* embryogenesis. Rather than examining the transcriptomes of groups of staged embryos, the authors perform RNA-Seq on single embryos. Altogether, they examined the transcriptomes of 122 different embryos spanning a 170 min interval of development, from 10 minutes to 3 hrs following fertilization. This encompasses the period of rapid nuclear division cycles and waves of zygotic genome activation. The authors then use computational models to produce pseudotime profiles of gene activities.

The reported method seems to be quite sensitive, enabling detection of the first zygotic transcripts at early stages, beginning at nc7. The authors both confirmed previous studies on the timing of zygotic activation and extended the datasets by identifying additional genes that are active during the minor wave of zygotic transcription during nc8, 9, and 10. This information will certainly be of value to the *Drosophila* research community.

There is a particular emphasis on sex-specific profiles of gene activity. The authors confirm earlier reports of the activation of two key sex determinants during the minor wave of zygotic genome activation, *scute* and *SisA*. As expected, these genes are expressed prior to the onset of *Sxl*. The authors also identify a novel gene that exhibits early sex-specific expression, *CG14427*, although its function remains unknown.

In summary, the use of single embryo sequencing and computational deconvolution of temporal dynamics is clear and clever. The method is simple and should be readily adopted by researchers in the field.

Authors' response to the first round of review

Reviewer #1

Major concerns

1. The first matter, which comes up at several places throughout the manuscript, is the idea that hand staging of embryos is difficult and time-consuming. While somewhat time-consuming, staging embryos visually is not that difficult, indeed numerous undergraduate researchers I work with have been trained to do so consistently and reliably. The earlier experiments that were mentioned for incorporating erroneously staged embryos were performed at a time where proper replication was not possible due to the costs of sequencing, and so data from one or two embryos were used (for example, the cited Lott et al., 2011 had only one replicate for some timepoints as each embryo required an entire lane of Illumina sequencing at that time), now proper replication is much more affordable.

Response: We agree with the reviewer that proper replication for RNA-

sequencing experiments has become much more affordable, and the larger number of replicates available makes errors in staging less relevant, due to more effective identification of statistical variation caused by contamination from embryos of the wrong stage. We have found that some trainees take weeks to stage embryos consistently and mistakes can only be identified by performing qPCR on select genes. If samples are excluded from sequencing, therefore, it is often based on an arbitrary cutoff of gene expression levels. Also, replicates with a significant amount of contamination can still skew the results and systemic mistakes are hard to identify. Overall, our idea is to completely avoid hand staging to make experiments easier and more reproducible. To accommodate the reviewers' concerns, however, we have toned down the manuscript text in several places (line 74, line 141 to 145).

While the method proposed in this manuscript is clearly the way that things are headed, it may be that the cost of sequencing close to two hundred embryos is prohibitive for some researchers, and hand staging a small number of replicates is more accessible. The authors might include some discussion about the number of samples needed for their method here, and the continued precipitous drop in sequencing costs could be used to motivate their method as the best one for the future.

We thank the reviewer for this comment. The method we are using is a miniaturized version that significantly reduces library preparation costs to approximately \$20 per sample. Also, in our experience a sequencing depth of 6 M reads per embryo is sufficient bringing the total costs in our hands to \$36 per embryo. To incentivize more researchers to use our method we have now included some more specific discussion about the number of samples and sequencing depth necessary (lines 394-402).

2. In a couple of places in the text (detailed below), I have concerns related to how the method used to remove unfertilized eggs may affect their results. As reported (line 106), the authors use expression of previously reported early transcribed genes to remove unfertilized eggs from inclusion in their datasets. Presumably, then, these genes represent early zygotic transcription. - In line 118, the authors claim that their pseudo-time starts at 10 minutes after fertilization. As the method of removing unfertilized eggs also removes all timepoints before any zygotic transcription begins (when maternal transcripts entirely control development), it's difficult to understand how this can represent 10 minutes after fertilization, as it does not include any of the time before any zygotic transcription is present. Do

the authors mean 10 minutes after the earliest zygotic transcription, instead? Please clarify.

Response: We thank the reviewer for this comment. We think that the reviewer's concerns are mostly due to a lack of clarity and can be addressed with more concise language. In our hands, it takes approximately 10 minutes to collect and freeze embryo samples. During this period, we recognize that embryos continue to develop. As such, the youngest embryo in our data set is at least 10 minutes into development. We clarified this in the text in lines 133-135. The reviewer is correct that our method can only identify unfertilized eggs once zygotic transcription starts. At the beginning of development, mRNA decay is the dominate process for Response to Reviewers determining the pseudo-temporal ordering of samples in our method, with many more genes down- than up-regulated, 291 and 66 respectively, between clusters 1 and 2. Decay of certain maternally deposited transcripts occurs even in unfertilized eggs once they are activated, but only fertilized eggs will start transcribing genes. This allows us to identify and exclude samples that, due to mRNA decay patterns, are assigned a high pseudo-time but lack evidence of early expressed genes, such as *scw*, *sc*, and *esg*. For example, if in general embryos are found to continuously increase their expression of *esg* after pseudo-time 20, we expect that all embryos after this timepoint will indeed express *esg*. If we find an embryo sample at pseudo-time 30 that is not expressing *esg*, we can conclude that that sample is an unfertilized egg. This can be visually observed in Supplemental Figure S1. We have revised the text and clarified this in line 118 to line 127.

- For the mRNA decay of maternally deposited transcripts section (line 235), again I have the same concern as above that the method used to remove unfertilized eggs necessarily includes some genes with zygotic transcription. While presumably only a small proportion of transcripts are zygotic at their first timepoint, if the point is to identify maternal genes that are not zygotically transcribed in order to examine the degradation of these transcripts, this is not a clean dataset to use for this purpose. I appreciate that the authors compare their data to an older dataset that is better designed for this purpose (Thomsen et al., 2010). I would suggest that the authors acknowledge this limitation in their dataset when describing this analysis.

Response: We again apologize for the lack of clarity. We acknowledge that our method measures total mRNA levels and does not distinguish between maternal and zygotic mRNA. For this reason, we limited our analysis of decay

processes to before the onset of major zygote genome activation and included only transcripts which are maternally deposited. We added the following sentence to clarify this limitation. “While our method is unable to distinguish maternal mRNA from zygotic mRNA, it can be used to study mRNA decay of maternally deposited transcripts before the zygote takes complete control of its own transcription” (line 260 to 263).

Minor concerns: 1. Line 250 - was there thought to be some relationship between the amount of an mRNA maternally deposited into the egg and the rate of degradation of that particular mRNA? I'm not sure mechanistically how that would work given what is known about degradation mechanisms. Perhaps it's interesting then to provide counter-evidence, as is done here.

Response: Thank you. This analysis was performed as a proof-of-principle to show that it is possible to use this method to study mRNA decay of maternally deposited transcripts before major ZGA. It not only shows that RNA abundance does not impact decay rates, but also that our dataset is not skewed towards low abundant transcripts. We have now included a statement in the manuscript to clarify this point (line 283-286).

2. Line 295 - Lott et al., 2011 shows that X chromosomal dosage compensation is not fully active by the end of NC 14, just that MSL-2 transcripts are present in males at a high level by what this study terms NC14C. This study posits a second, non-MSL mechanism that is active for some X-linked genes prior to MSL activation. Functional studies (such as Strukov et al., 2011, PLOS Biology) show likely onset of MSL-mediated dosage compensation around gastrulation/cycle 15.

Response: We thank the reviewer for pointing out this oversight. We clarified the contributions of the Lott et al. 2011 publication and included Strukov et al. 2011 as a reference. (lines 325-328)

3. Starting on line 396- discussion of runt. I think the discussion might be confusing the role of runt as an activator of Sxl with the potential role of runt as a Sxl target, which may be involved with the multiple roles of runt during this developmental timeperiod (i.e. earlier as an XSE and later as a pair rule gene). Despite being an XSE runt is known to be relatively dosage insensitive, and recent evidence (Mahadeveraju, Jung, and Erickson, 2020, G3) showed that runt helps maintain expression from the establishment promoter of Sxl in female embryos. I would suggest that the authors revisit this section of the text as I think these findings (and more generally runt as a regulator of Sxl) help explain their observations.

Response: We thank the reviewer for pointing out this complication. We have revised this section to clarify the interpretation of the data (line 436 to 442).

4. Starting on line 542 - Fixation, staining, and staging of embryos. I couldn't find in the manuscript why embryos were fixed and stained? A number of the previous single-embryo methods that involve staging do not fix or stain the embryos. Please clarify for what datasets these methods were used, and for what purpose.

Response: We apologize for the confusion. This passage refers to the qPCR experiment that was used on pooled samples of hand staged embryos to determine the precise time of transcription onset for early transcribed genes that were identified using our pseudo-time approach. We have clarified this in the text (line 208, and in the methods section in line 584).

Reviewer #2

Major concerns:

1. The novelty and all major findings of the current study are depending on the accurate organization of individual embryos along the pseudo-time. So, it is very important to ensure the trajectory is correct. The authors used different dimensionality reduction methods and showed these embryos form a linear organization. This is a good sign, but it doesn't guarantee all embryos are aligned in the same order. I suggest the Authors to use different trajectory analysis approaches, that have been developed and used for single cell RNAseq studies, such as Monocle 3 and Velocity for RNA velocity, to validate and correct (if any wrong) for the pseudo-time analysis.

Response: We agree with the reviewer that the correct organization of individual embryos along the pseudo-time is very important to our study. We used StemID2, an algorithm for the inference of differentiation trajectories developed to perform lineage analysis on single cell sequencing data. Lineage tree projection coordinates are used to calculate a pseudo-temporal order by StemID. To categorically validate this computationally derived pseudo-time, we used 3 different previously published datasets (Lott et al. 2011; Sandler and Stathopoulos 2016; Kwasnieski et al. 2019) (Figure 2G-I), in addition to our own, using manually staged embryos (Supplemental Figure S3 A-F). We have modified the text in this section of the manuscript to include more details of our approach (line 113 to 116 and line 153 to 154). We decided to focus on

only one bioinformatic tool in the manuscript to provide the reader with a defined, easy to use analysis pipeline. RaceID/StemID allows researchers with minimal bioinformatic knowledge to perform the analysis. During the initial analysis of our data, we tested and compared different bioinformatic tools (including Monocle) to infer the developmental trajectory and determine the pseudo-temporal order. During the earliest time points, within cluster 1, the different tools assigned different pseudo-time positions to the embryos because there are a few detectable transcriptome changes, as this is before zygotic transcription or maternal mRNA degradation are initiated. In subsequent clusters however, the ordering of embryos is extremely consistent with only very minor differences in the pseudo-time position. Comparing the ordering produced by Monocle3 and our method, we get at most a 4-position difference in the pseudo-time order between the two methods.

2. Most analyses are focused on validating previous findings, but not on newly identified factors. It will be nice to show new findings from current study. For example, how many new zygotic genes are identified? Which factors' activation timings should be updated? Showing such information will benefit the development community.

Response: A differential gene expression analysis between clusters in our dataset is presented in Figure 3. Fold change and p-values for all genes are provided in Supplemental Table S3 for each comparison. Due to the difference in approach used in our study (continuous time scale) compared with published studies (categorical time scale), a direct comparison is difficult. Also, we decided not to focus on the identification of new zygotically expressed factors, which collectively have been reported in previous studies (Lott et al. 2011; Kwasnieski et al. 2019; De Renzis et al. 2007), rather, the strength of this study is to provide the accurate point in development at which transcription for a specific gene is initiated and the dynamic nature of the transcriptional profile at unprecedented temporal resolution. As such, we report a unique resource for the scientific community that will allow anyone who is interested in doing so to determine the expression pattern for a gene of interest in great temporal detail. In addition, with this new methodology in hand, researchers can now assess the impact of genetic or environmental perturbations on early development without the need for manual staging.

3. For figure 3 and 4, the Authors performed the comparison analysis in the cluster level. Within each cluster, is there significant embryo-embryo variation? Will this variation impact the cluster level comparison?

Response: Variation within each cluster was accounted for during multiple steps in our analysis (Supplemental Table S6, source code). First, the method utilized unsupervised clustering to group only samples of a certain similarity; if an embryo is too different from the rest, it will cluster by itself. Secondly, using RaceID, robustness of the clusters was assessed in various ways, including determining Jaccard's similarity, which in our analysis is consistently greater than 0.9 for all clusters. Finally, the RaceID workflow includes a step to identify outliers. For this study, we used the default setting of findoutliers (probthr = 0.001). The details for RaceID, StemID and FateID can be found in (Herman et al. 2018).

Minor points:

1. In *Drosophila*, XXY represents a female, XY represents a male. By sex-chromosome genotyping analysis, how to distinguish the gender? Did the authors remove embryos of XXY genotype?

Response: To account for the rare case of having XXY or other sex chromosome abnormalities, we amplified both the X and the Y-chromosome by qPCR for each embryo. Those embryos with a similar Ct value in both the X and Y-chromosome were classified as males (XY) while embryos amplifying only the X were classified as females (XX). In our experiments, we did not find embryos with Ct values higher in the X-chromosome compared to the Y-chromosome, suggesting a XXY genotype.

2. It will be nice to show quality control data such as # UMI and # genes per embryo in Supplement.

Response: We have now included that information in supplemental Table S5, referred to in line 104 in the main text and line 564 in the methods section.

3. For Fig 3, it is not clear how Zld dependent, H2A.Z positive or negative, are defined. Please add details in the Methods.

Response: We have added details about how Zld dependent, H2A.Z positive or negative are defined in the methods section (line 640-651).

4. For Volcano plots in Fig 3, in the legend padj was used, but p-value was used in the plots.

Response: We thank the reviewer for this observation. The P-value label was used erroneously. As such, we have replaced P-value with the correct padj value in the volcano plots.

5. In line 539: "Only results with a false discovery rate (FDR) ≤ 0.5 are shown." Should it be 0.05.

Response: Again, we thank the reviewer for catching this error. We have revised the sentence to state that “Only results with a false discovery rate (FDR) ≤ 0.05 are shown”.

Reviewer #3

Perez-Mojica et al. present a new method for analyzing temporal dynamics of gene expression during *Drosophila* embryogenesis. Rather than examining the transcriptomes of groups of staged embryos, the authors perform RNA-Seq on single embryos. Altogether, they examined the transcriptomes of 122 different embryos spanning a 170 min interval of development, from 10 minutes to 3 hrs following fertilization. This encompasses the period of rapid nuclear division cycles and waves of zygotic genome activation. The authors then use computational models to produce pseudotime profiles of gene activities. The reported method seems to be quite sensitive, enabling detection of the first zygotic transcripts at early stages, beginning at nc7. The authors both confirmed previous studies on the timing of zygotic activation and extended the datasets by identifying additional genes that are active during the minor wave of zygotic transcription during nc8, 9, and 10. This information will certainly be of value to the *Drosophila* research community. There is a particular emphasis on sex-specific profiles of gene activity. The authors confirm earlier reports of the activation of two key sex determinants during the minor wave of zygotic genome activation, *scute* and *SisA*. As expected, these genes are expressed prior to the onset of *Sxl*. The authors also identify a novel gene that exhibits early sex-specific expression, CG14427, although its function remains unknown. In summary, the use of single embryo sequencing and computational deconvolution of temporal dynamics is clear and clever. The method is simple and should be readily adopted by researchers in the field.

Response: We thank the reviewer for their supportive comments and for noting our meticulous approach to this study.

Referees' report, second round of review

Reviewer 2

The authors have addressed all my concerns, especially the one about the

pseudo time analysis, which is critical for the key findings in this paper. The authors should be congratulated for a nice work!

Authors' response to the second round of review

N/a